# Human Milk: Fast Determination of Docosahexaenoic Acid (DHA)

**Mahyara Markievicz Mancio Kus-Yamashita** [1,2,*]**, Cristiane Bonaldi Cano** [2]**, Vânia Claudia Barros Monteiro** [3] and **Regina Maria Catarino** [4]

1    School of Pharmaceutical Science, University of São Paulo—USP, São Paulo 05508-000, Brazil
2    Chemistry, Physics and Sensory Core, Food Centre, Adolfo Lutz Institute, São Paulo 05508-000, Brazil
3    Medical Assistance to the State Public Servant Institute (IAMSPE), São Paulo 05508-000, Brazil
4    Pathology Centre, Adolfo Lutz Institute, São Paulo 05508-000, Brazil
*    Correspondence: mahyara.kus@ial.sp.gov.br

**Abstract:** Human milk provides all the nutrients required by babies during the first six months of their life. Human milk lipids represent the main source of energy, contributing almost 50% of the total energy content. Additionally, fatty acids ensure the correct development of children in the prenatal, postnatal, and infant phases. Docosahexaenoic acid (DHA) is essential for visual and cognitive development, and its presence during childhood can affect long-term health. This study aimed to optimize and validate a methodology for the direct determination of DHA in human milk. We used 20 samples of human milk from lactating women living in the city of Itu, São Paulo, who attended Basic Health Units from September 2019 to September 2020, and a sample of certified reference material from the National Institute of Standards and Technology. The proposed methodology consists of a validated mixed transesterification process without prior lipid extraction, optimized by factorial design. This methodology can be successfully used in human milk samples as it is both precise and accurate. The values of DHA in the sampled milks were similar to those in European countries and lower than those in Asian countries due to diet.

**Keywords:** human milk; polyunsaturated acid; breastfeeding; gas chromatography; chemometric approach

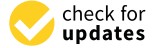



## 1. Introduction

Human milk (HM) is considered the ideal food as it is the best source of nutrients for infants, and it has been associated with several short- and long-term health benefits, being tailored to infants' needs [1,2].

Evidence indicates that breastfeeding can attenuate late metabolic diseases and protect against obesity and type 2 diabetes [2]. Fat in HM is the second largest macronutrient and plays the most important role in providing nutrients to babies, contributing to almost 50% of the total energy content [3].

The composition of human milk is influenced by the nutritional status, quantity, and quality of the mother's diet. The presence of certain vitamins and minerals can be impacted by malnourishment, overconsumption of fatty foods, or alcohol intake, with consequences for the nutrients in human milk [4].

HM fat is composed of fatty acids (FAs), which ensure the correct development of children during the prenatal, postnatal, and infant phases. FAs are responsible for the energy processes that occur in cells and are the main building material of cell membranes and precursors of important metabolic compounds such as prostacyclins, prostaglandins, thromboxanes, and leukotrienes [5]. Palmitic (16:0), oleic (18:0 n-9), and linoleic (18:2 n-6) acids are the main FAs present in HM [6].

Docosahexaenoic fatty acid (DHA) has 22 carbon atoms and 6 double bonds (22:6 n-3). Over the last 20 years, there has been concern over the amount of DHA during pregnancy and lactation, since it represents more than 10% of brain fat and is essential for

child brain development [4]. DHA is the main polyunsaturated fatty acid in the human brain and retinal rods. It is therefore essential for fetal brain and retina development during pregnancy and for the further development of the baby [7].

DHA plays a major role in psychomotor neurodevelopment in the first few months of life and is supplied in large amounts by HM during this phase [7]. In the first 6 months of life, when exclusive breastfeeding is recommended, the infant brain doubles in weight. The brain to body weight ratio in infants (0.1) is greater than that of adults (0.02), so nutrient deficits are of greater concern. Much of the increase in brain weight is attributed to an increase in gray matter, corresponding to the formation of neural synapses rich in DHA [4]. The benefits of DHA for the fetus and baby are supported by extensive literature confirming the importance of adequate omega-3 (n-3) intake for maternal health (to reduce the risk of preterm birth and postpartum depression), HM composition, and the general health of babies [7]. Although the human body has an enzymatic pathway necessary for the synthesis of DHA (the n-3 fatty acid with the longest carbon chain and the highest degree of unsaturation) from the metabolic precursor $\alpha$-linolenic acid (ALA, C18:3 n-3), there is clear experimental evidence that the conversion of ALA to longer-chain FAs is insufficient to ensure adequate tissue levels. The conversion efficiency of ALA to eicosapentaenoic acid (EPA) has been shown to be highly variable (less than 10%), and the conversion of ALA to DHA has even lower conversion efficiency [7,8].

The FA composition of HM has been studied in different countries worldwide, and the average level of DHA in mature breast milk varies between $0.32 \pm 0.22\%$ of total FAs [9]. Marine mammals have high DHA levels as it can be synthesized from aquatic phytoplankton and transferred through the food chain. Fatty fish are a rich dietary source of DHA. Dietary intake of DHA is low in many parts of the world, which has been linked to low fish consumption. In an analysis of worldwide DHA levels in breast milk, four of the top five locations reporting the highest concentrations are from coastal or island populations with a diet rich in seafood. In contrast, those with the lowest reported levels are inland or developed countries with less seafood in their diets [9,10].

Higher levels of DHA in breast milk (exceeding 1% of total FAs) are observed in coastal populations and are associated with the consumption of marine foods [11].

Common methods for determining FAs in HM involve several steps, such as lipid extraction with an organic solvent and derivatization for gas chromatography, which involve long and complex procedures. Errors and contamination can occur during solvent extraction, evaporation, and purification. Therefore, methods that do not include prior extraction of lipids with organic solvents, including only lipid extraction and FA derivatization, reduce the time required, usage of solvents, and losses that can generate errors in the procedure [12].

Currently, direct transesterification methods without prior fat extraction are commonly used. These methods mix the sample with esterification reagents such as methanol hydrogen chloride; methanol acetyl chloride; methanol sodium methoxide; methanol sodium hydroxide (NaOH), hydrochloric acid and methanol solution; and sulfuric methanol acid and boron trifluoride solution (BF3). This saves time and resources as the lipid extraction and derivatization steps occur the same time [13].

Scientists have used direct methods to prepare FA methyl esters (FAME) based on the original method proposed by Lepage and Roy (1986) [14]. Cruz-Hernandez et al. [15] observed that the direct method can be used for fatty acid quantification in human milk. Other authors demonstrated similar results in other areas: for samples of marine animals [16]; when analyzing egg yolks [17]; for dairy products [18]; and for milk powder and conjugated linoleic acid supplementation [19]. In 2007, the American Oil Chemists' Society (AOCS) [20] published the Ce 2b-11 method of direct methylation of lipids using alkaline hydrolysis with methanolic NaOH, organic solvent extraction (n-hexane), and methylation with BF3. This method, however, is not applicable for milk fats and marine oils or oils with long-chain polyunsaturated fatty acids (LC-PUFAs) or micro-encapsulated oils. For such cases, the AOCS recommends the use of the Ce 2c-11 method [20].

Optimization achieved through experimental designs can be used to evaluate critical factors and facilitate the interpretation of results. This type of approach has several advantages, including the reduction of experiments, analysis time, and the use of samples and reagents. Although a two-level full factorial design is commonly used to assess significant effects in an experiment, when there are many factors, a fractional factorial design is more appropriate for reducing the number of experiments [21].

This study aimed to optimize and validate a methodology for the determination of DHA in HM by the direct method, which is based on transesterification using mixed catalysis to simplify the analysis of FAs in HM, thus extending the knowledge of FAs secreted by the mammary glands.

## 2. Materials and Methods

### 2.1. Materials

#### 2.1.1. Sample

A sample of HM composed of a mixture of 5 HMs was used to optimize the methodology; a sample of certified reference material of infant formula from the National Institute of Standards and Technology (NIST) 1849 was used to confirm the methodology.

The HM samples used in this study were obtained from research conducted with the approval of the Ethics Committee in Research with Human Beings of the Nossa Senhora do Patrocínio University Center, Itu, São Paulo. The study was conducted at the Basic Health Units in the city of Itu, São Paulo and involved 20 nursing mothers in good health, aged between 18 and 45 years, who were approached in their first consultation. Breastfeeding women willing to sign an informed consent form, who were mothers with an only child, had full-term deliveries, who exclusively or predominantly breastfed, were at low risk, and willing to provide a sample of breast milk were included. This research was approved by the Ethics Committee for Research with Human Beings of the Nossa Senhora do Patrocínio University Center, Itu, São Paulo, Brazil (Approval No. 3.523.211).

Milk was manually expressed immediately after breastfeeding using techniques recommended by the National Network of HM Banks [22]. Milk samples (10 mL) were collected in 50 mL polypropylene bottles. All the samples were properly identified, immediately cooled, and packed in an isothermal box with reusable ice. Subsequently, they were frozen in a freezer ($-18\,°C$), transported to the Adolfo Lutz Institute, and frozen until analysis.

#### 2.1.2. Reagents, Standards

We used the following analytical-grade reagents: ammonium hydroxide, sodium chloride, and sodium hydroxide and the following chromatographic-grade reagents: methanol and n-hexane. FA methyl ester C23 (99% purity) was used as an IS, and a 10 mg mL$^{-1}$ solution of DHA methyl ester was used for identification.

### 2.2. Methods

For the analysis of DHA, FAs were directly extracted and derivatized based on the methodology proposed by Hartman and Lago (1973) [23] and adapted by Maia and Rodrigues-Amaya (1991) [24]. Approximately 1.6 g of breast milk was added to a glass tube with a lid, followed by 1 mL of a 2.5 mg mL$^{-1}$ solution of methyl ester C 23 (IS) and 8 mL of a 2 mol L$^{-1}$ solution of sodium hydroxide in methanolic medium. Subsequently, the mixture was vortexed for 30 s and heated in a water bath at 70 °C for 5 min. After cooling, 10 mL of esterifying solution (ammonium chloride, methanol, and sulfuric acid) was added, vortexed for 30 s, and heated in a water bath at 70 °C for 5 min. The glass tube was kept at 20–25 °C until cooling, after which 3 mL of saturated sodium chloride solution and 3 mL of n-hexane were added. The upper part was transferred to a vial and analyzed using gas chromatography.

The methodology was optimized according to the following parameters: sample mass (X1), volume of solutions used (X2), and heating time in a water bath at 70 °C (X3) through a Fractional Factorial Design 23-1 with replication [25]. The following levels were studied:

1.6 g and 5.0 g for X1; 4/5 mL and 8/10 mL for X2; and 5 min and 15 min for X3. For the analysis of the factorial design, the concentration of DHA in g $100\,g^{-1}$ in HM was calculated based on the methodology proposed by Kus et al. (2009) [26] and using Statistica® software.

The methodology was validated according to the parameters recommended by the National Institute of Metrology Standardization and Industrial Quality (Instituto Nacional de Metrologia, Qualidade e Tecnologia—Inmetro) Validation Guide [27], namely: matrix effect, selectivity, linearity, quantification limit, detention limit, sensitivity, accuracy (recovery), and precision (repeatability and intermediate precision). Additionally, a certified reference material (CRM) was used to evaluate the accuracy and precision of the optimized method.

The transesterified samples were analyzed by gas chromatography using a Thermo Focus GC flame ionization detector (FID) and a Thermo Triplus automatic injector. The components were separated in a fused silica capillary column with an Agilent HP-88 bis-cyanopropyl siloxane stationary phase (100 m, 0.25 mm, 0.25 μm). The chromatographic conditions were based on a study by Fournier et al. (2007) [28], namely: injector temperature: 250 °C; detector temperature: 300 °C; pressure: 200 kPa; oven temperature: 60 °C for 5 min; $15\,°C\,min^{-1}$ to 165 °C (1 min); $2\,°C\,min^{-1}$ to 225 °C (17 min), flow: $1.90\,mL\,min^{-1}$, drag gas: hydrogen (flow: $30\,mL\,min^{-1}$); makeup gas (nitrogen): 30 mL/min; synthetic airflow: 300 mL/min; column pressure: 175 kPa; dividing ratio of the 1:50 sample. The inject mode was splitless.

A standard solution of FAME with 37 components was used to cope with all these conditions, and all compounds were separated.

DHA was quantified by internal standardization, using the IS of the FAME C23:0 and correction factors of the experimental FID in relation to the IS [20,29,30]. Equations were used, and calculations were performed according to Kus et al. (2009) [26].

## 3. Results and Discussion

### 3.1. Optimization of the Direct Method for Quantification of DHA in Human Milk

The critical parameters involved in direct transesterification are listed according to previous tests and their influence on the esterification reaction kinetics [31], namely, sample mass, solution volume, and heating time, which are independent variables. Optimization was performed through the fractional factorial design of a $2^{3-1}$ model with replication. The DHA concentrations (dependent variable) obtained by chromatographic analysis were used in the factorial design analysis using Statistica® software.

Analysis of variance (Table 1) and Pareto chart (Figure 1) were used to assess the effect of variables on the response of the model generated by factorial design. Table 1 and Figure 1 show that the variables are significant, as their values are <0.05, considering a 95% confidence level. Mass was the variable that contributed less to the model, having a negative effect, which means that smaller mass values resulted in greater yields in the reaction. The volume of the solutions, in turn, was the second most important variable, and its effect was positive; that is, a greater volume of reagents resulted in a better yield in the transesterification reaction. Although heating time was a significant variable, its impact was smaller than that of the previously mentioned variables, as shown in Figure 1a, because of its slight significance, which does not affect the reaction conditions by the levels chosen in the factorial design.

The correlation coefficient (R) was used to assess the generated model, as it shows the goodness-of-fit of the data to the suggested model (in this case, the linear model) and measures the amount of variance around the mean explained by the model. This value was 0.9968 (Table 1). A coefficient close to 1 indicates a good relationship between the experimental data and the suggested model. The predicted and observed values for the concentration of DHA, which are shown in Figure 1b, were also used to assess the model. Most of the values were dispersed in a narrow range close to the straight line, which indicated a good correlation between the predicted and observed responses, a good fit of the proposed linear model, and little dispersion between the replications.

**Table 1.** Analysis of variance—ANOVA—for the model generated through fractional factorial design $2^{3-1}$ with replication.

| R-Sqrt = 0.99686; Adj: 0.99769 2 ** (3-1) Design; MS Residual = 0 | | | | | |
|---|---|---|---|---|---|
| ANOVA | | | | | |
| Factor | SS | df | MS | F | P |
| (1) sample mass | 0.00015 | 1 | 0.000015 | 2157.745 | 0.000001 |
| (2) volume of solutions | 0.000006 | 1 | 0.000006 | 818.748 | 0.000009 |
| (3) time | 0.000000 | 1 | 0.000000 | 56.039 | 0.001703 |
| Error | 0.000000 | 4 | | | |
| Total SS | 0.000022 | 7 | | | |

SS: sum of squares, df: degrees of freedom; MS: mean squares; F: F test; P: confidence level. **: fractional factorial design.

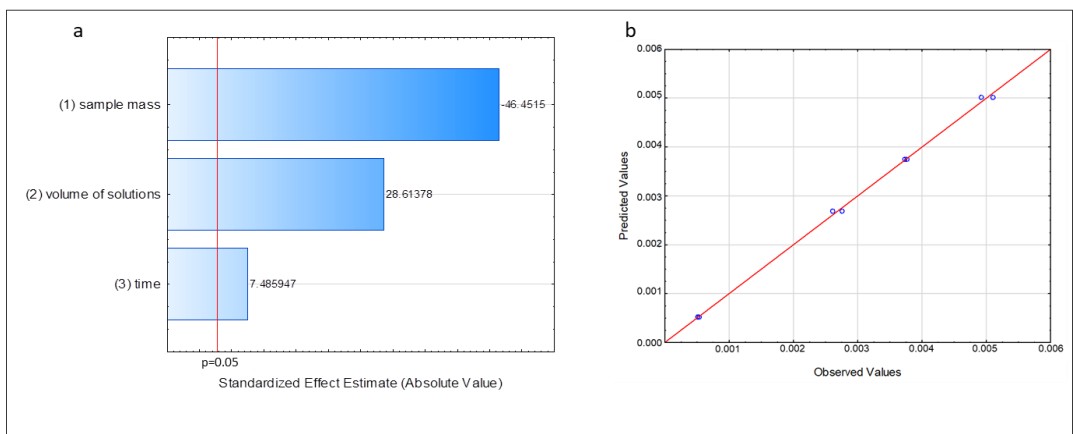

**Figure 1.** Figures obtained from the model generated by fractional factorial design $2^{3-1}$. (**a**) Pareto chart. (**b**) Predicted and observed values for the concentration of DHA. Source: Statistica®.

A response surface chart was constructed to determine the optimal values for each variable and investigate the interactions between them (Figure 2). These charts show the effects of the independent variables and the interaction of each variable with the response variable [25,32]. Figure 2 shows the response surface chart constructed by setting the mass variable at level –1, owing to its relevant contribution to the transesterification reaction, as shown in Figure 1a. Thus, there is a group of optimal values (red color) for the variable volume of reagents and heating times (Figure 2).

Figure 2 suggests that the volume of the solutions and sample mass are correlated, regardless of the heating time; therefore, the impact of the heating time on the reaction yield is low, and any of the levels can be defined by the interval proposed in this study. Therefore, a heating time of 5 min was chosen to reduce the duration of the experiment, and the following conditions were defined for the method: mass: approximately 1.6 g; solutions, 8 mL of methanolic 0.5 N NaOH, and 10 mL of esterifying solution; heating time, 5 min. These values are similar to those reported by Sendzeliene et al. (2004) [31], who observed that the rate of the esterification reaction depends on the initial mass of the sample and that the higher the concentration of FAs, the greater the reaction rate in relation to the reaction time. Compared to longer reaction times, shorter reaction times resulted in a greater conversion rate of FAs to methyl esters. These behaviors were also observed by Farag et al. (2011) [33].

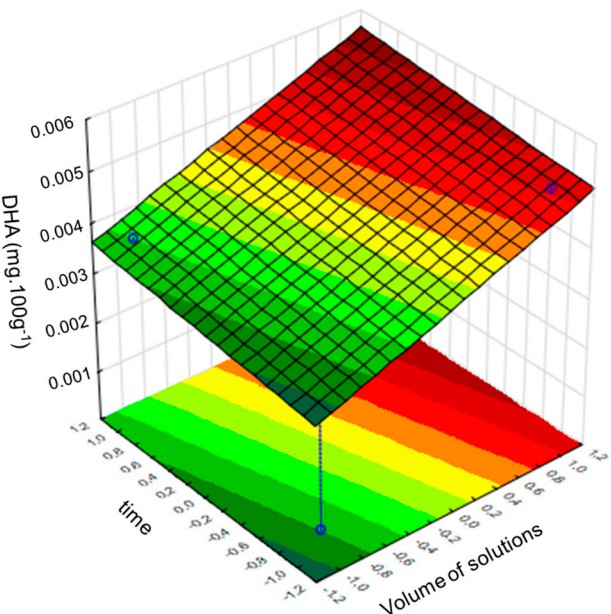

**Figure 2.** Response surface for the variables: volume of solutions and heating time with the sample mass fixed at level −1. Source: Statistica®.

### 3.2. Validation of the Direct Method for the Quantification of DHA in Human Milk

The validation parameters were based on the Inmetro Validation Guide [27] and Association of Official Analytical Chemists [29]. We assessed the selectivity and matrix effect for the HM at concentrations of 0.022, 0.066, and 0.110 mg mL$^{-1}$. The DHA values were calculated by the F test and paired *t*-test (0.9857, 0.2285, and 0.5568, respectively, for each concentration), considering a significance level of 95%. The tests revealed that there was no matrix interference; therefore, it had no effect on the precision and result in the analyte.

In the linearity study, seven concentration levels were evaluated, with three replicas at each level. No outliers were detected by the Grubbs test, with a confidence limit of 95%, which shows that the data are homoscedastic [27]. The correlation coefficients of the areas corresponding to the concentration levels ranged from 0.19 to 6.1%. The residuals of the analytical curves exhibited random behavior. The analytical curves were also considered linear by the analysis of variance of the regression and F test, following the Inmetro Guide [27] and Ribeiro et al. (2008) [21]. Furthermore, the value of r$^2$ = 0.99999, expressed in Table 2, reveals a good fit of the data to the linear model. The data from these analytical curves, detection limits, quantification, and recoveries are presented in Table 2. The analytical curve is shown in Figure 3. The quantification and detection limits were calculated using an analytical curve [27]. The linear range was from 0.020 to 0.141 mg mL$^{-1}$ DHA, including the DHA concentrations obtained in HM studies.

**Table 2.** Validation parameters for quantification of docosahexaenoic acid in human milk.

| Parameters | | Values |
|---|---|---|
| Regression Equation (Linear model) | | $1.06 \times 10^7 x + 4.05 \times 10^6$ |
| r$^2$ | | 0.9999 |
| LD (95%) | | 0.015 mg mL$^{-1}$ |
| LQ (95%) | | 0.020 mg mL$^{-1}$ |
| Recovery (%) | L1 = 0.022 mg mL$^{-1}$ | 99.10% |
| | L2 = 0.065 mg mL$^{-1}$ | 101.33% |
| | L3 = 0.109 mg mL$^{-1}$ | 101.15% |

LD = detection limit; LQ = quantification limit; L1, L2, L3 = levels.

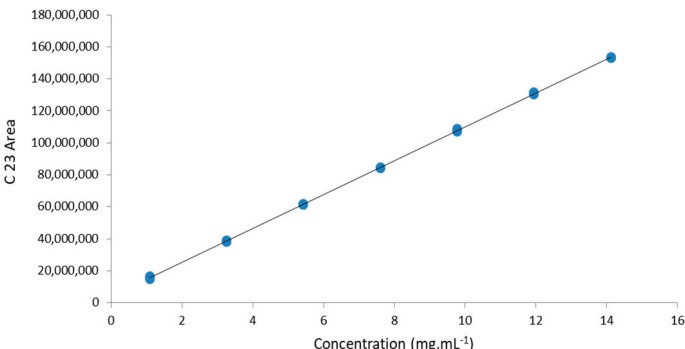

**Figure 3.** Analytical curve of the internal standard C23.

The recovery values presented in Table 2 were in accordance with the recommendations of the Inmetro Validation Guide [27], that is, between 97 and 103%, which is a narrower range of acceptance than the range from 92 to 105% recommended by the AOAC (2013) [29]. Both consider that DHA values are in a proportion of 1% in the analytical curve.

Precision and intermediate precision tests were performed, and the relative standard deviation for HM were 1.48% and 0.94%, respectively. These values are in accordance with Inmetro, with a repeatability limit of 2.7% for a range of 1% of the analyte [27] and are in accordance with AOAC (2013) [29], being within the limit of 2.0% at the same concentration level.

As a certified sample of HM was not available, the data obtained in the validation were confirmed by tests conducted on a certified reference sample of NIST infant formula (NIST 1849) in its powder form with approximately 30% lipids. The formula was reconstituted in distilled water at a concentration of 10% to obtain a lipid value close to that of HM (2–3%). Figure 4a shows the DHA values obtained for the reconstituted samples from the NIST. Seven tests were performed, and all were within the range allowed by the certificate (0.0179 ± 0.0024 g 100 g$^{-1}$).

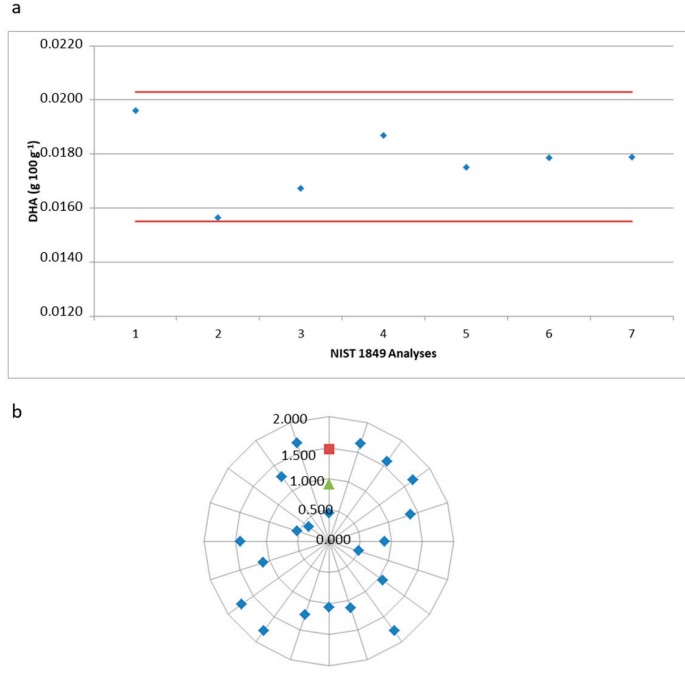

**Figure 4.** Reproducibility and precision of the optimized methodology for docosahexaenoic acid (DHA) analysis. (**a**) Docosahexaenoic acid—DHA values (g 100 g$^{-1}$) in the NIST 1849 sample. (**b**) Relative standard deviation of human milk samples (blue), precision (red), and intermediate precision (green).

Direct methodology works for the determination of DHA in human milk are very rare. Cruz-Hernandez et al. (2013) [15] proposed a direct methodology for the determination of AG in HM, for DHA and obtained a correlation coefficient of 5.1% for precision and 6.94% for intermediate precision. There are no official AOAC or AOCS methods for determining DHA in HM. Therefore, in order to compare the performance of official methods with the method proposed here, data obtained from DHA determination in cow's milk and/or infant formula were used, as these have lipid and DHA levels similar to HM and consequently provide the closest matrices. Li, Kotoshi, and Srigley (2019) [34] analyzed cow's milk with additional omega 3, following the direct methodology laid out by AOCS Ce 2c-11, and observed a correlation coefficient of 7.7% (with 6 replicas). Golay and Moulin (2016) [35] published a collaborative study of the direct method AOAC 2012.13 and determined a correlation coefficient for DHA in infant formula and milk between 5.47 and 14.64%. The performance parameters verified for the determination of DHA in HM by our group therefore agree with or are smaller than those of other authors and official methods for matrices similar to the HM, suggesting that this alternative method can be applied routinely in the laboratory.

Therefore, according to the data obtained in accordance with the validation parameters recommended by the Inmetro Validation Guide and the standard deviation values obtained for the CRM samples, the direct method proposed in this work is innovative, as it uses only one mixed transesterification step for the analysis of fatty acids, without the previous extraction of the fat. Therefore, it is suitable for the quantification of DHA in HM, for demonstrating great accuracy and precision, using lesser reagent, being faster, having lower cost, and being simpler to operate; therefore, it can be considered an alternative method for DHA determination in HM. This method is promising for the quantification of other fatty acids present in HM and can be studied for other food matrices.

### 3.3. Applicability of the Method in Real Samples

Twenty milk samples from nursing mothers living in the city of Itu, São Paulo, were analyzed to demonstrate the applicability of the direct method for determining DHA in HM; the mean, standard deviation, minimum, and maximum values are presented in Table 3 and Figure 4b shows the relative standard deviation values of the 20 samples in this study, the precision, and the intermediate precision of validation. A chromatogram of HM in the DHA elution region is illustrated in Figure 5.

**Table 3.** Docosahexaenoic acid (DHA) values milk from nursing mothers living in the city of Itu, São Paulo.

| Sample (ID) | DHA (g 100 g$^{-1}$) |
|:---:|:---:|
| 1 | 0.0435 |
| 2 | 0.0824 |
| 3 | 0.0783 |
| 4 | 0.0721 |
| 5 | 0.0356 |
| 6 | 0.1018 |
| 7 | 0.0986 |
| 8 | 0.0900 |
| 9 | 0.2717 |
| 10 | 0.0415 |
| 11 | 0.0260 |
| 12 | 0.1145 |
| 13 | 0.0327 |

**Table 3.** *Cont.*

| Sample (ID) | DHA (g 100 g$^{-1}$) |
|---|---|
| 14 | 0.0582 |
| 15 | 0.0576 |
| 16 | 0.0455 |
| 17 | 0.0597 |
| 18 | 0.1393 |
| 19 | 0.0787 |
| 20 | 0.0630 |
| Average | 0.0795 |
| Standard deviation | 0.0539 |
| Minimum | 0.0260 |
| Maximum | 0.2717 |

ID = sample identification.

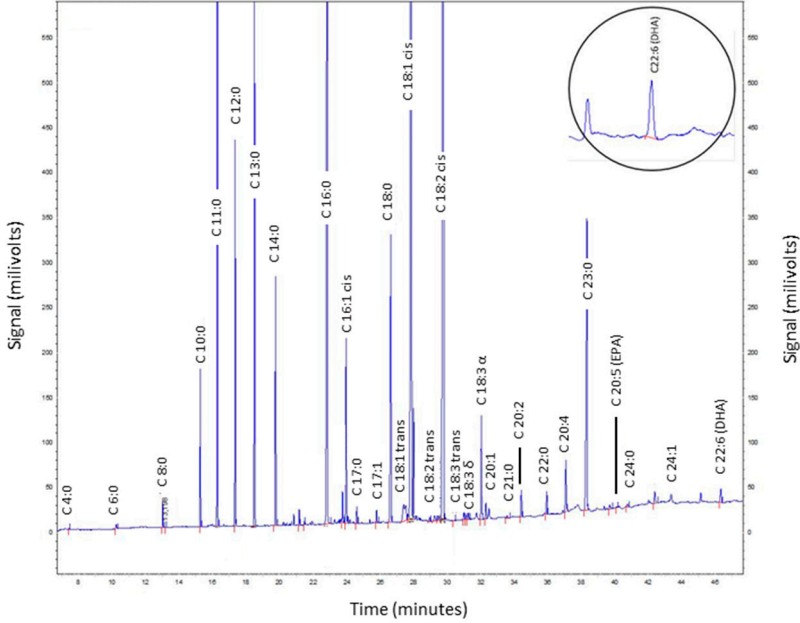

**Figure 5.** Chromatogram of a human milk sample (ID 5). The DHA elution region is highlighted.

The relative standard deviation values of DHA found in most HM samples, ranging from 0.5 to 1.5%, as shown in Figure 3b, demonstrated that they were accurate according to the limit recommended by the Inmetro Validation Guide of 2.7% (1% of the analyte) [27] and were within the limit of 2.0% (1% of the analyte) recommended by the AOAC (2013) [29]. Figure 4 shows the limits established by the author of the reference material (NIST) and the analyses performed in our study, therefore demonstrating the accuracy of the methodology and that our study is in accordance with the established criteria.

The mean DHA content in HM observed in our study was 0.0795 g 100 g$^{-1}$, i.e., 0.26% in FAs (transition milk). The values obtained in the present study were close to those observed in Italy (0.28–0.35%), Spain (0.31–0.38%), and Nordic countries, such as Iceland (0.30%) and Denmark (0.35%) [36]. However, other studies found relatively higher values, as in the case of a review of 55 studies that gathered 4374 samples of HM from several countries and found 0.46% DHA in FAs in 957 transition milk samples, 0.51% DHA in FAs in 943 HM colostrum, and 0.31% DHA in FAs in 2397 mature HM samples [37]. A recent study conducted by Guiffrida et al. (2022) [38] with 223 mothers from Europe found

DHA values ranging from 0.34 to 0.40% in FAs, similar to those reported by Floris et al. (2020) [37]. Ueno et al. (2020) [39], when evaluating Japanese women, showed a higher amount of DHA in HM compared to those revealed in our study and the European study, with a mean of 0.55% DHA for women who never ingested a DHA supplement and 0.74% for those supplemented. These higher values are due to the source of DHA in their diet, as in Japan, the consumption of fish and algae is higher than that in other places [39].

Studies on DHA supplementation in pregnant or lactating women have shown an increase in the concentration of DHA in HM. A Dutch study by Goor et al. (2009) [40] found a 43% increase in DHA in HM after 2 weeks of supplementation with 200 mg of DHA, with a total value of 0.60% of DHA. Similar results were found by Sherry and Marriage (2015) [4] in the United States: supplementing infants with 200 mg of DHA increased DHA content by 26% in LH after 6 weeks, with DHA levels rising from 0.23% to 0.36%. the authors also evaluated DHA levels after supplementation with 400 mg of DHA, and observed an increase of 154%, from 0.18% to 0.46%. Valencia-Naranjo et al. (2022) [41], in a Colombian pilot study, found that DHA values in HM increased from 0.19% to 0.29% after 3 months of DHA supplementation. Jensen et al. (2000) [42], in the United States, had similar results, with an increase from 0.27% to 0.44% DHA after supplementation. Dunstan et al. (2007) [43] supplemented DHA in pregnant women after 20 weeks of gestation and observed the amount of DHA in the HM after 3 days and 6 weeks of breastfeeding. The control group had a DHA value of 0.5% and 0.25% after 3 days and 6 weeks, respectively, while the supplemented group showed increased values of 1.15% and 0.42% for the same time points. A review by Amaral et al. (2017) [44] noted that in 22 studies with DHA supplementation in pregnant or lactating women, all demonstrated a strong correlation of DHA supplementation and increased DHA content in HM.

Therefore, the value of DHA in HM varies across studies, and this may be related to maternal nutrition, as it is the source of FA for the composition of HM [36]. In some cases, DHA supplementation can affect the FA composition in mature HM from the third trimester of pregnancy [39].

### 4. Conclusions

After optimizing the studied variables (lower mass, greater reagent volume, and shorter heating time), the direct method based on mixed transesterification and without prior fat extraction proved to be precise and accurate (validation study) for the analysis of DHA in HM. The method could be successfully used in real samples of HM, showing low values of relative standard deviations, and is promising for other FAs and food matrices. The DHA values found in our study are like those in European studies and lower than those in Japanese studies, as food interferes with the amount of DHA in the HM. Data on DHA in HM are scarce and may contribute to the nutritional adequacy of nursing mothers. Due to the impact of DHA supplementation in pregnant and lactating women, it is important to establish an alternative, faster, and more reliable method for determining DHA in HM.

**Author Contributions:** M.M.M.K.-Y.: investigation, formal analysis, validation, writing—original draft; V.C.B.M.: conceptualization, investigation, writing—reviewing; C.B.C.: investigation, data curation, writing—reviewing; R.M.C.: conceptualization, resources, funding acquisition, writing—reviewing. All authors have read and agreed to the published version of the manuscript.

**Funding:** This research was funded Coordination for the Improvement of Higher Education Personnel (CAPES—Brazil) (Proc. No. 88882.444208/2019-1).

**Data Availability Statement:** Available online: https://docs.google.com/spreadsheets/d/1PYbWI1ULUib-b39r4UxKMdkuSPn_LNca/edit?usp=sharing&ouid=11198306744898624131 5&rtpof=true&sd=true (accessed on 28 December 2022).

**Acknowledgments:** The authors are grateful for the partnership in project of Adolfo Lutz Institute (IAL) and for the financial support provided by Coordination for the Improvement of Higher Education Personnel (CAPES).

**Conflicts of Interest:** The authors declare no conflict of interest.

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
