# Peer review of "Human Milk: Fast Determination of Docosahexaenoic Acid (DHA)"

_analytica, doi:10.3390/analytica4010006_

Round 1

Reviewer 1 Report

Line 23- other? if samples were collected in Sao Paulo (not in Europe)

Introduction have to be improved 

Line 49 reference? 

Line 67 "Some Scientists" reference? 

Line 84 "different"??? women

Please add example chromatogram in Results section 

Line 182: R? but what was dependent and what independent value? In this study PLS should be used not coefficient of determination 

Table 2: It is regression equation not analytical curve 

Table 3 Some results are above linearity range

In discussion section it worth to compare obtained results with patients after dietary supplementation. here you have example paper: Journal of lipid research, 41, 2000, 1376-1383 

Author Response

Line 23- other? if samples were collected in Sao Paulo (not in Europe)

In text.

Introduction have to be improved  - In text

Line 49 reference? 

In text.

Line 67 "Some Scientists" reference? 

In text.

Line 84 "different"??? women

The word "different" has been removed from the text. Explaining: a pool of 5 breast milk with different amounts of DHA was made.

Please add example chromatogram in Results section  

In text

Line 182: R? but what was dependent and what independent value? In this study PLS should be used not coefficient of determination 

In text

Table 2: It is regression equation not analytical curve 

In text

Table 3 Some results are above linearity range

Samples that are above the range of linearity studied were previously diluted.

In discussion section it worth to compare obtained results with patients after dietary supplementation. here you have example paper: Journal of lipid research, 41, 2000, 1376-1383 

In text

Reviewer 2 Report

The manuscript “Human milk: fast determination of docosahexaenoic acid (DHA)” is an interesting work aimed to improve a methodology (proposed by Hartman and Lago in 1973) for the direct determination of DHA in human milk. The author's research is well-established; they have provided necessary studies in optimization, validation and real sample tests, but still some modifications are needed before it can be accepted for publication.

1.     Please correct the formatting of Figure 1 and enlarge the font.

2.     Table 1 used undefined abbreviations. Please revise.

3.     Please clarify the method used to determine the Analytical Curve, LD and LQ.

4.     Proper citation is required for Figure 3.

5.     Please compare the performance of the proposed/optimized method with current commercial methods/approaches/techniques and literatures.

6.     What the meaning and significance of “standardized effect estimate (absolute value)” in figure 1a?

7.     How many replicates were tested for each concentration level of DHA in the analytical curve? And what is the standard deviation of replicates?

8.     Please provide figure(s) of the analytical curve in the main content.

9.     Please provide parameters and settings of the gas chromatography and provide the Chromatogram(s).

Author Response

  1. Please correct the formatting of Figure 1 and enlarge the font.

In text

  1. Table 1 used undefined abbreviations. Please revise.

In text

  1. Please clarify the method used to determine the Analytical Curve, LD and LQ.

       The analytical curve was determined according to the Inmetro Guide and Eurachen Guide 2014. LD and LQ based on the Inmetro guide and item 4.2.2.2 of the ICH Validation Guide:  Validation Of Analytical Procedures Q2(R2).

  1. Please compare the performance of the proposed/optimized method with current commercial methods/approaches/techniques and literatures.

      In text

  1. What the meaning and significance of “standardized effect estimate (absolute value)” in figure 1a?

      The standardized estimated effect reflects the contribution of each variable in the generated model, with a significance level of 95%, as well as its correlation with the studied response, which can be synergistic (positive effect) or antagonistic (negative effect).

  1. How many replicates were tested for each concentration level of DHA in the analytical curve? And what is the standard deviation of replicates?

     In text

  1. Please provide figure(s) of the analytical curve in the main content.

     In text

  1. Please provide parameters and settings of the gas chromatography and provide the Chromatogram(s).

    In text

Round 2

Reviewer 1 Report

The article was revised accordingly reviewer comments, without statistics. My recommendation was to apply PLS (partial least squares) and the Authors used a correlation coefficient (R). 

Author Response

Dear reviewer, in the attached file you will find our considerations about the statistical analysis.

Reviewer 2 Report

The authors have made detailed elaboration and reasonable modifications to the problems I mentioned before. After re-reading the revised manuscript, I think it has a reasonable structure, clear logic and sufficient argumentation, which can be published.

Author Response

Dear reviewer, thank you for your considerations, they have greatly enriched the text of the article.